# A systematic review to identify research priority setting in Black and minority ethnic health and evaluate their processes

Halima Iqbal[1,2]*, Jane West[2], Melanie Haith-Cooper[1], Rosemary R. C. McEachan[2]

1 Faculty of Health Studies, University of Bradford, Bradford, United Kingdom, 2 Bradford Institute for Health Research, Bradford Teaching Hospital NHS Foundation Trust, Bradford, United Kingdom

* h.iqbal23@bradford.ac.uk

**Data Availability Statement:** All relevant data are within the manuscript and its Supporting Information files.

## Abstract

### Background

Black, Asian and minority ethnic communities suffer from disproportionately poorer health than the general population. This issue has been recently exemplified by the large numbers of infection rates and deaths caused by covid-19 in BAME populations. Future research has the potential to improve health outcomes for these groups. High quality research priority setting is crucial to effectively consider the needs of the most vulnerable groups of the population.

### Objective

The purpose of this systematic review is to identify existing research priority studies conducted for BAME health and to determine the extent to which they followed good practice principles for research priority setting.

### Method

Included studies were identified by searching Medline, Cinnahl, PsychINFO, Psychology and Behavioral Sciences Collection, as well as searches in grey literature. Search terms included "research priority setting", "research prioritisation", "research agenda", "Black and minority ethnic", "ethnic group". Studies were included if they identified or elicited research priorities for BAME health and if they outlined a process of conducting a research prioritisation exercise. A checklist of Nine Common Themes of Good Practice in research priority setting was used as a methodological framework to evaluate the research priority processes of each study.

### Results

Out of 1514 citations initially obtained, 17 studies were included in the final synthesis. Topic areas for their research prioritisation exercise included suicide prevention, knee surgery, mental health, preterm birth, and child obesity. Public and patient involvement was included in eleven studies. Methods of research prioritisation included workshops, Delphi techniques,

**Funding:** This work was supported by the National Institute for Health Research (NIHR) under its Applied Research Collaboration (ARC) Yorkshire and Humber in the form of Ph.D. funding to HI [NIHR200166], the UK Prevention Research Partnership (UKPRP) in the form of funding to JW and RM [MR/S037527/1], the NIHR Clinical Research Network in the form of funding to JW, and the NIHR ARC Yorkshire and Humber in the form of funding to RM.

**Competing interests:** The authors have declared that no competing interests exist.

surveys, focus groups and interviews. The quality of empirical evidence was diverse. None of the exercises followed all good practice principles as outlined in the checklist. Areas that were lacking in particular were: the lack of a comprehensive approach to guide the process; limited use of criteria to guide discussion around priorities; unequal or no representation from ethnic minorities, and poor evaluation of their own processes.

## Conclusions

Research priority setting practices were found to mostly not follow good practice guidelines which aim to ensure rigour in priority setting activities and support the inclusion of BAME communities in establishing the research agenda. Research is unlikely to deliver useful findings that can support relevant research and positive change for BAME communities unless they fulfil areas of good practice such as inclusivity of key stakeholders' input, planning for implementation of identified priorities, criteria for deciding on priorities, and evaluation of their processes in research priority setting.

## Introduction

Current evidence demonstrates disproportionately poorer health outcomes for Black and Minority Ethnic (BAME) groups. In particular, the prevalence of Type 2 diabetes is reported to be as much as six times higher in UK South Asians compared to Europeans [1] and disparities in mental health care for BAME groups represent a serious public health concern [2] with a significantly disproportionate number of people from BAME backgrounds detained under UK mental health legislation in hospitals in England and Wales [3].

The extent and seriousness of disparities in health has been further demonstrated in the recent global pandemic of severe acute respiratory syndrome caused by covid-19 and which has disproportionately affected vulnerable and marginalised populations such as BAME groups [4] being up to twice as likely to die from covid-19 in the UK than people of White British ethnicity [5]. However, it is worth noting that this is a new pandemic and inconsistent and emerging findings continue to be reported. This pandemic has exposed the severe extent of existing socioeconomic health and structural inequalities, ranging from poverty to barriers to accessing care, and crowded living conditions [6] among these groups that have been exacerbated by covid-19.

Research priority setting has the potential to reduce disparities in health by making research more efficient in solving health problems. Involving the local population addresses the issue of equity and attends to the needs of the most vulnerable groups within the population, while reinforcing the links between research, action and policy [7]. There is no consensus on the definition of research priority setting but there is agreement on a range of activities that centre on identifying, prioritising and reaching agreement on the research areas or questions deemed important to stakeholders [8]. Historically, researchers and funders have generally set healthcare agendas themselves [9]. More recently, it has been recognised that research needs to address questions that are relevant to the people it intends to make a difference for, give them a voice [10] and mitigate waste [8]. Key stakeholders include healthcare professionals, policy makers, patients, and their families, as well as the public more generally. These questions should be answered using the most appropriate methods, and research results need to be reported in a manner that is comprehensive, transparent, and accessible [11].

The past decade has seen an increase in research priority setting exercises in a range of areas [12] and there are increasing efforts to identify shared research priorities using explicit processes [13]. In their narrative review of health research priority setting methods, models and frameworks, Bryant et al. (2014) found that among eleven different priority setting exercises identified from the United Kingdom, Australia, the United States and Canada, none had been evaluated to check their prioritisation processes or assess the extent to which the exercise had achieved its objectives [14]. It is also unclear whether research prioritisation exercises have been undertaken for BAME health. This systematic review is interested in identifying whether there has been any progress in research priority process evaluation since then, with a specific focus on BAME health, given that BAME communities suffer worse health outcomes than the wider population [15], the stark increase in research priority setting in the past decade [12], along with an increase of discourse around evaluation of research priority setting initiatives [13].

Therefore, this study aimed to identify and evaluate existing research priority setting studies conducted for BAME health. Applying a critical lens to their processes may inform ways to improve future research prioritisation for BAME populations and increase the value and contribution of research aimed to improve the health of BAME communities.

## Study questions

1. Are there health research priority setting studies conducted for BAME health?

2. Have they adhered to good practice principles in health research priority setting?

## Methods

The systematic review followed the standards of the Preferred Reporting Items for Systematic Reviews and Meta-Analysis (PRISMA) statement [16] (S1 File). The search was undertaken between July 6th-7th, 2020, in four electronic health databases: CINAHL, MEDLINE, PsychINFO and PBSC. The following Boolean search term combinations were used:

i. "research priority setting" [all fields] OR "research prioritization" [all fields] OR "research prioritisation" [all fields] OR "research priorities" [all fields] OR "research agenda" [all fields] AND

ii. "Black and minority ethnic" [all fields] OR "BAME" [all fields] OR "ethnic group" [all fields] OR "ethnic groups" [all fields] OR "minority groups" [all fields] OR "multicultural" [all fields] OR "Asians" [all fields] OR "immigrants" [all fields] OR "indigenous" [all fields] OR "Aborigines" [all fields]

To ensure the full scope of published literature within each database was targeted, we searched databases from their inception to July 2020. Titles and abstracts published in English only were included due to time limitations. Abstracts were screened for relevance. Given its limited timeline, the principal researcher (HI) independently conducted the article search. Studies were included in the full text screening that used a qualitative, quantitative and mixed method design. Searches in the grey literature included: reference lists of included articles, Google Scholar, Cochrane methods priority setting, and the James Lind Alliance (a well-established priority setting partnership method). The search string 'research priority setting and Black and minority ethnic health' was applied to Google Scholar. The first ten pages of Google were examined for eligible articles. All authors contributed and refined the review's search strategy. Two authors (HI and MC) applied the critical appraisal criteria.

## Inclusion and exclusion criteria

The review was developed to include any study that outlined a process of conducting a research prioritisation exercise. Studies must outline the characteristics of the participants, discuss the methods used to obtain research and identified well-established outcomes. Studies that made no mention of health research or did not describe the research prioritisation process were excluded. Studies were included if they focused on obtaining research priorities specifically for BAME populations. This includes studies that sought to identify research priorities for ethnically mixed populations and involved them in the process, provided that differences in priorities from BAME groups and the wider population were described. See Table 1 for the inclusion and exclusion criteria.

To ensure credibility of the process, all authors discussed and agreed the selected papers. References were managed with EndNote X9 for ease. After removing duplicates, HI independently screened the title and abstract of 1,080 records including three found in grey literature. In total, 32 studies were selected for full text examination. The PRISMA flowchart, including reasoning for study exclusions, are displayed in Fig 1.

## Quality appraisal tool

Each of the identified studies were assessed using a quality appraisal tool specifically designed for health research priority setting. In the absence of a gold standard approach to research prioritisation, a checklist of nine common themes for good practice in health research priority setting was [17] used to determine whether the research priority setting exercises from the studies adhered to good practice principles in their processes as reported by the checklist. This checklist has been used previously to evaluate research priority setting exercises [13, 18–20], and has effectively identified weaknesses prevalent in research prioritisation exercises. As this tool was specifically designed with health research prioritisation in mind, it could identify issues that may otherwise have been overlooked by traditional quality appraisal tools.

The checklist is organised into three domains which were used to critically appraise the studies: *preparatory work*, *deciding on priorities*, and *after priorities have been set*. Each domain contains corresponding practices that further identify the goals in each step. Within *preparatory work*, there are five related practices: context, use of a comprehensive approach, inclusiveness, information gathering, and planning for implementation; within *deciding on priorities*: criteria, and methods for deciding on priorities; and within *after priorities have been set*: evaluation and transparency. See Table 2 for a detailed description of each theme.

**Table 1. Inclusion and exclusion criteria.**

| Included | Excluded |
|---|---|
| Studies that directly elicited and identified research priorities (e.g., topics or questions) for BAME health | Studies assessing priorities for practice and policy (quality indicators) |
| Studies must outline a process of research priority setting, including participants characteristics, study type and an outcome | Non-research articles (policy documents, clinical guidelines, commentaries, editorials) |
| Studies that sought to identify research priorities in White and BAME populations and involved the public in the exercise must display the differences in identified priorities between both groups | Studies that involved White and BAME groups in identifying their health priorities yet did not discuss disparities between priorities |
| UK and international studies | Study protocols |
| Studies written in the English language only | Conference reports, workshop or meeting that failed to include information about the participants and methods |
| | Interventions to improve BAME health |
| | Priority setting exercises that were non-health research priority focused |

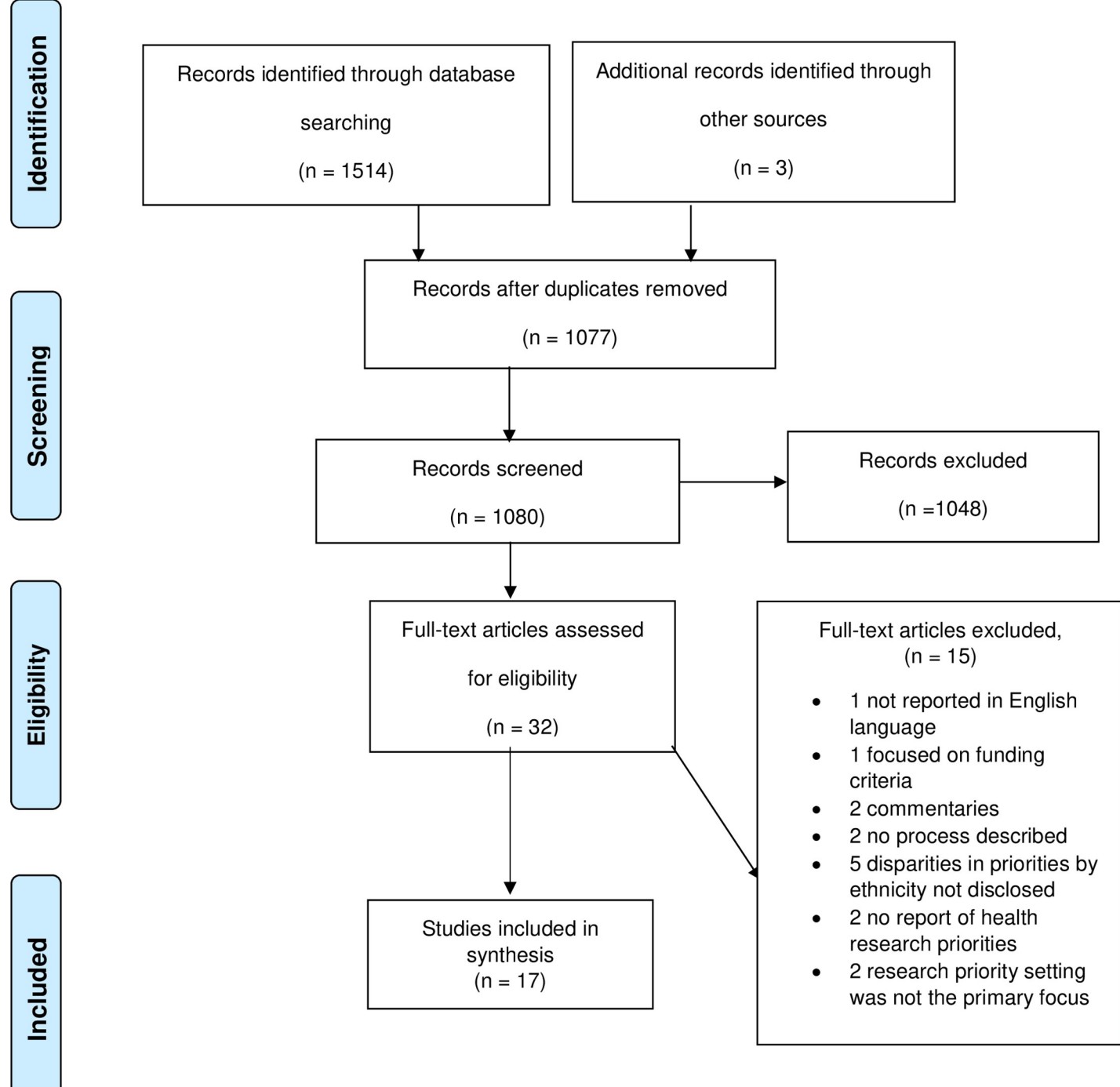

**Fig 1. PRISMA flow diagram.**

## Data collection process

We conducted a descriptive synthesis to summarize the study characteristics and outcomes, as well as how well each study matched up against the good practice principles as reported by the checklist. A quality score was assigned to each study and was based on the number of 20 good

Table 2. Checklist for health research priority setting [17].

| Theme | Description |
| --- | --- |
| **Preparatory work** | |
| 1—Context | 1 The resources available for the exercise were reported<br>2 The focus of the exercise was clearly stated (what it was about and who it was for)<br>3 The underlying values or principles were clear<br>4 The health environment in which the process took place was described<br>5 The research environment in which the process took place was described<br>6 The political environment in which the process took place was described<br>7 The economic/financial environment in which the process took place was described |
| 2—Use of a comprehensive approach | 8 The process of priority setting was described in detail |
| 3—Inclusiveness | 9 The participants involved in setting research priorities were described<br>10 An appropriate representation of expertise was included<br>11 An appropriate representation of sex was included<br>12 An appropriate representation of regional participation was included<br>13 Relevant health sectors and other constituencies were included |
| 4—Information gathering | 14 The information and sources used to inform the priority setting exercise were referenced |
| 5—Planning for implementation | 15 Plans for translation of research priorities were discussed<br>16 Who will implement the research priorities and how? |
| **Deciding on priorities** | |
| 6—Criteria | 17 Relevant criteria to focus discussion on setting priorities were stated |
| 7—Methods for deciding on priorities | 18 Approach for deciding on priorities was described (e.g., consensus or metrics based) |
| **After priorities have been set** | |
| 8—Evaluation | 19 When and how evaluation of the established priorities and the priority setting process will take place were defined (e.g., multiple sessions) |
| 9—Transparency | 20 Clarity about the approach used, i.e., how priorities were set |

practice criteria it had met in Table 2. One researcher (HI) independently extracted study characteristics, methods, and outcomes. Comprehensive data extraction checklist forms for the quality synthesis were developed to extract the relevant data. Two researchers (HI and MC) extracted the data to quality appraise the studies.

## Results

We identified 17 studies meeting our inclusion criteria. These studies were conducted in research priority setting for BAME health in a range of different topic areas. Three studies were conducted for mixed ethnic groups [21–23]. All 17 studies can be seen in Table 3. The topic areas were child obesity [21, 24], mental health [25], suicide [26, 27], cancer [28], E-health [29], knee replacement surgery [30], pre-term birth [31], healthy school development [32], and in health more generally [22–24, 33–36]. The prioritisation exercises were conducted to determine health research priorities in these topic areas for a range of different ethnicities in high income countries. They concerned Latino health in the US [24, 37], refugee and immigrant health in the US [25, 26, 32], the health of indigenous Australians [28, 29, 36], the health of native Americans [27], the health of Asian youth in New Zealand [34, 35], South Asian health in the UK [30, 33], the health of minority and underrepresented communities in the US [22, 23], and Black and Hispanic health in the US [21, 31]. Six studies did not include patient and public involvement (PPI) as participants in establishing research priorities [21, 24, 25, 27, 32, 35], Five studies had PPI involvement alongside other stakeholders such as healthcare professionals, academics, researchers, and decision makers [26, 28, 30, 33, 37]. Four studies

**Table 3. Study characteristics for the included empirical studies with quality score.**

| Study ID | Country | Title | Topic and scope | Population included in the Identification of priorities | Method | Main outcome (research Priorities) | Quality score (Based on met criteria in the checklist–see Table 2) |
|---|---|---|---|---|---|---|---|
| Flores et al., 2002 [24] | USA | The health of Latino children: urgent priorities, unanswered questions and a research agenda | General health | Paediatricians, health researchers, academic dentist, anthropologists, academic nurse, environmental health expert, dean of a school of public health No public involvement (total n = 13) | Workshop | Research agenda included: (1) greater inclusion of Latino children in medical research (2) analysis of study data by pertinent Latino subgroups (3) more research on Latino child health issues that can elucidate social and economic determinants of health and use of health services for all children, such as cross-border health and the healthy immigrant effect (4) enhancing early educational opportunities for Latino children (5) training healthcare professionals more extensively in cultural competency | 6/20 (30%) |
| Colucci et al., 2010 [25] | Australia | Setting research priorities in refugee mental health | Refugee mental health | Academics, key practitioners, and policy makers No public involvement (total n = 71) | 2 online surveys | Key research priorities included (1) the design and delivery and location of mental health services for refugee clients (2) how existing services can be adapted and extended for refugee clients (3) the prevalence of mental health problems in refugee clients (4) factors promoting resilience and successful transition to life in the new country of settlement | 7/20 (35%) |
| Colucci et al., 2017 [26] | Australia | A suicide research agenda for people from immigrant and refugee backgrounds | Suicide in immigrant and refugee populations | Policy makers, service providers, academics, service users, carer advocates (total n = 138) | Online Delphi with two rounds of questionnaire. | Greatest priority was given to: (1) access and engagement with suicide prevention services (2) suicide protective and risk factors compared to populations not from immigrant and refugee backgrounds (3) culturally appropriate assessment of suicide risk | 9/20 (45%) |
| Goold et al., 2017 [22] | USA | Priorities for patient-centered outcomes research: the views of minority and underserved communities | General health in minority and underserved communities | Academic and community partners (n = unknown) Members from minority and medically underserved communities White (63) Black or African American (98) Other (22) (total n = 183) | Interviews | Greatest priority was given to: (1) quality of life, (2) patient-doctor, (3) access, (4) special needs (5) compare approaches. | 12/20 (60%) |
| | | | | | Focus groups | Black participants were less likely to prioritize research on causes of disease, new approaches, and compare approaches than White participants. | |

*(Continued)*

**Table 3.** (Continued)

| Study ID | Country | Title | Topic and scope | Population included in the Identification of priorities | Method | Main outcome (research Priorities) | Quality score (Based on met criteria in the checklist–see Table 2) |
|---|---|---|---|---|---|---|---|
| Manikam et al., 2017 [33] | England | Using a co-production prioritization exercise involving South Asian children, young people and their families to identify health priorities requiring further research and public awareness | co-production of child health research and public awareness agendas | Heath care practitioners from a range of backgrounds (n = 27) South Asian children and families (n = 35) (total n = 62) | Systematic literature review Scoping survey Focus groups | Health care practitioners prioritized public awareness on obesity, mental health, healthcare access, vitamin D and routine health checks and research on nutrition, diabetes, health education and parenting methods. South Asians prioritized research into the effectiveness of alternative Medicines. Both healthcare practitioners and South Asians prioritized increased research or public awareness on mental health illness, blood and organ donation, obesity, and diet. | 12/20 (60%) |
| McNeely at al., 2017 [32] | USA | How schools can promote healthy development for newly arrived immigrant and refugee adolescents: research priorities | identification of research priorities for promoting the school success of immigrant and refugee youth | Researchers, service providers, educators, policymakers No public involvement (n = 132) | Modified CHRNI framework | Highest priority research options were: (1) evaluating newcomer programs identifying how family and community stressors affect newly arrived immigrant and refugee adolescents' functioning in school (2) identifying teachers' major stressors in working with this population (3) identifying how to engage immigrant and refugee families in their children's education | 13/20 (65%) |
| Morris., 2017 [28] | Australia | Identifying research priorities to improve cancer control for indigenous Australians | identify emerging research priorities in Indigenous cancer control. | Researchers, public health practitioners, advocacy groups, allied health and other related professionals, Indigenous cancer survivors and their families, and Indigenous community groups. (total n = 225) | Online survey | Identified research priorities included: (1) cancer prevention and early detection (2) health literacy (3) culturally appropriate care for Indigenous patients, survivors, and families. | 10/20 (50%) |

(*Continued*)

**Table 3.** (Continued)

| Study ID | Country | Title | Topic and scope | Population included in the Identification of priorities | Method | Main outcome (research Priorities) | Quality score (Based on met criteria in the checklist–see Table 2) |
|---|---|---|---|---|---|---|---|
| Franck et al., 2018 [31] | USA | A novel method for involving women of color at high risk for preterm birth in research priority setting | A research agenda for pre-term birth in women of colour | BAME women at high risk of preterm birth (total n = 12) | Novel RPAC framework Focus groups | A list of Top 10 research priorities including: (1) How does a mother's stress affect the baby? (2)-What are the most effective ways to improve patient-provider communication, particularly when patients perceive insensitive and rude comments from health care workers? (3) What is the most effective care for pregnancy and high-risk pregnancy? For example, if African American women are at higher risk, why isn't there specialized care to improve outcomes? (4) What causes Sudden Infant Death Syndrome? (5) Does the type of insurance you have determine the type of care that you get, or the quality of your care and is care different based on insurance status or race? | 12/20 (60%) |
| Ramirez., 2011 [37] | USA | Salud America! Developing a national Latino childhood obesity research agenda | To identify research priorities to address Latino childhood obesity | Academics, researchers, health educators, administrators, managers, clinicians, public health workers, students, community (total n = 313) | Modified three-round Web-based Delphi | 25 research priorities identified within the domains of society; community; school; family; individual. These include: Society: Policies that subsidize accessibility of healthy foods to improve diet among Latino families Community: built environment policies involving collaborations with multiple stakeholders to promote physical activity School: health, nutrition, and active physical education classes as part of the school curriculum Family: engaging Latino families as advocates of childhood obesity prevention initiatives at the community and school levels Individual: programs making physical activity more attractive than watching TV or playing video games | 13/20 (65%) |

*(Continued)*

**Table 3.** (*Continued*)

| Study ID | Country | Title | Topic and scope | Population included in the Identification of priorities | Method | Main outcome (research Priorities) | Quality score (Based on met criteria in the checklist–see Table 2) |
|---|---|---|---|---|---|---|---|
| Peiris-John et al., 2016 [35] | New Zealand | Stakeholders views on factors influencing the wellbeing and health sector engagement of young Asian New Zealanders | priorities on the health and wellbeing of Asian youth | Opinion leaders, key decision makers on Asian youth health from the academic field, in health service planning and community organisations No public involvement (total n = 6) | interviews | Key priority themes identified were: (1) cultural identity (2) integration and acculturation (3) barriers to help-seeking | 5/20 (25%) |
| Maar et al., 2010 [29] | Canada | Reaching agreement for an Aboriginal e-health research agenda: the Aboriginal telehealth knowledge circle consensus method | Develop an Aboriginal E-health research agenda | Aboriginal council members; Aboriginal community; regional and provincial, and federal leaders, and policy makers (total n = 40) | Novel Aboriginal telehealth knowledge consensus method containing 7 cycles | Priorities fell into 6 distinct topics (1) ethical principles for Aboriginal e-health Research (2) internet-based national information for Aboriginal e-health initiatives; (3) research related to e-health education and professional development; (4) sustainability; (5) best practices; and (6) broader applications and impact of e-health on Aboriginal culture and communities. | 12/20 (60%) |
| Goold et al. 2018 [23] | USA | Members of minority and underserved communities set priorities for health research | General health in minority and underserved communities | Academic and community partners (n = unknown Members from minority and medically underserved communities 45% White ethnic 30% Black African American 8% Hispanic 6% Native American 4%Arab American, Arab, or Chaldean (total n = 519) | Focus groups Surveys | Highest priority was given to: child health research and mental health research. Other prioritised topics were Aging, access, promote health, healthy environment, and what causes disease Black/African American participants were less likely to prioritise mental health research Native American and Arab American participants prioritised research on culture and beliefs | 11/20 (55%) |
| Spurling et al., 2017 [36] | Australia | 'I'm not sure it paints an honest picture of where my health's at'- identifying community health and research priorities based on health assessments within an Aboriginal and Torres Strait Islander community: a qualitative study | Identify health priorities of the Aboriginal and Torres Strait Islander communities that could be translated into research themes | Key informants from an urban Aboriginal and Torres Strait Islander community (total n = 21) | Interviews | Three themes emerged, to be translated into research priorities (1) complex, interrelated intergenerational nature of health involving social, cultural, and environmental determinants of health (2) ambivalence about health assessments (3) community strength | 10/20 (50%) |

(*Continued*)

**Table 3.** (Continued)

| Study ID | Country | Title | Topic and scope | Population included in the Identification of priorities | Method | Main outcome (research Priorities) | Quality score (Based on met criteria in the checklist–see Table 2) |
|---|---|---|---|---|---|---|---|
| Wexler et al., 2015 [27] | USA | Framing health matters: advancing suicide prevention research with rural American Indian and Alaska native populations | Suicide research priorities in indigenous populations | Suicide researchers No public involvement (n = unspecified) | 3-day consensus workshop | Two main themes: (1) Expansive commitments of indigenous approaches to inquiry: holistic perspectives; focus on the past as well as the present and future (2) Community-level factors: conceptualising suicide as a social problem: localizing indigenous suicide rates in specific community contexts; development of community capacity and collaboration on design of local programs; emphasis on protective factors, resilience, and well = being | 7/20 (35%) |
| Bryan et al., 2020 [30] | Canada | A research agenda to improve patients' experience of knee replacement surgery: a patient-oriented modified Delphi study of South Asian origin in British Coumbia | Identify a research agenda for South Asian patients who undergo knee replacement surgery | South Asian patients and caregivers, healthcare professionals (total n = 53) | Focus groups Modified Delphi survey | A list of 25 priorities Top priorities both for patients and caregivers and for clinicians were (1) promoting exercise following surgery and (2) self-management after hospital discharge. One of the highest ranked topics for patients and caregivers was improving knee implants. Patients and caregivers prioritized research on promotion of exercise and self-management following surgery and improvement in knee implants. | 10/20 (50%) |
| Gallagher et al., 2010 [21] | USA | Identifying interdisciplinary research priorities to prevent and treat pediatric obesity in New York city | Child obesity in underrepresented Black and Hispanic communities | Obesity experts among different faculties at Columbia university including clinical scientists, clinicians, educators, service providers, and public health researchers No public involvement (total n = 55) | 4 focus groups Survey | A list of top 10 priorities including: (1) Integration of behavioural and cultural components into research (2) Contribution of health disparities on rates of childhood obesity in our communities (3) Social determinants of health and identification of previously unmeasured factors (4) Effectiveness of behavioural approaches targeted toward families (5) Translating current evidence into practice in the clinic and community activity | 12/20 (60%) |

(*Continued*)

**Table 3.** (Continued)

| Study ID | Country | Title | Topic and scope | Population included in the Identification of priorities | Method | Main outcome (research Priorities) | Quality score (Based on met criteria in the checklist–see Table 2) |
|---|---|---|---|---|---|---|---|
| Wong et al., 2015 [34] | New Zealand | Priorities for Asian youth health: perspectives of young Asian New Zealanders | identify priority areas for research on Asian youth health | Asian youth (total n = 15) | Focus groups | Themes identified were: (1) cultural differences and identity (2) racism and discrimination (3) access mental health issues | 4/20 (20%) |

E-health is the health services and information delivered through the internet and related technologies

involved only BAME PPI as participants [29, 31, 34, 36] and 2 studies involved multi-ethnic PPI in identifying priorities and highlighted the differences in identified priorities by ethnicity [22, 23]. The most common methods used to identify priorities were surveys, focus groups, interviews, Delphi techniques, and workshops.

The main outcomes of studies were the identification of a range of research priorities related to the topic area. The research priorities were expressed as prioritised research topics, priority themes, top 10 prioritised lists, and more extensive lists of research questions. Top identified priorities included the need for greater inclusion of Latino children in medical research [24], the design and delivery and location of mental health services for refugees [25], access and engagement with suicide prevention services for people from immigrant and refugee backgrounds [26], Expansive commitments of indigenous approaches to inquiry for suicide in indigenous populations [27], cancer prevention and early detection in indigenous Australians [28], ethical principles for aboriginal e-health research [29], research on quality of life for minority and underserved populations [22], child health research and mental health for minority and underserved populations [23], complex, intergenerational nature of health involving social, cultural, and environmental determinants of health for Aboriginal and Torres strait islander communities [36], research into the effectiveness of alternative medicines for South Asian children [33], promoting exercise following surgery for South Asian patients undergoing knee replacement surgery [30], evaluating newcomer programs identifying how family and community stressors affect newly arrived immigrants and refugee adolescents functioning in school [32], how does a mothers stress affect the baby? for women of colour [31], policies that subsidise accessibility of healthy foods to improve diet among Latino families [37], integration of behavioural and cultural components into research to prevent and treat paediatric obesity in predominantly Black and Hispanic communities [21], cultural differences and identity for young Asian New Zealanders [34] and cultural identity, integration and acculturation in young Asian new Zealanders [35].

## Assessing study quality against the checklist of good practice in research priority setting

None of the studies fulfilled all good practice principles as proposed by the checklist (see Table 4).

**Theme 1: Context.** Every study reported some contextual factors. All studies made the focus of their exercise clear and overall, studies reported the values and principles behind their exercise. However, only three studies explicitly included information regarding the resources

**Table 4. Appraisal of comprehensiveness of reporting.**

| Item | Study | Total studies *n* (%) |
|---|---|---|
| **Context** | | |
| *1-The resources available for the exercise were reported* | [22, 23, 31] | 3 (17.6) |
| *2-The focus of the exercise was clearly stated (what it was about and who it was for)* | [21–37] | 17 (100) |
| *3-The underlying values or principles were clear* | [21, 22, 24–37] | 16 (94.1) |
| *4-The health environment in which the process took place was described* | [21, 24–33, 35–37] | 14 (82.3) |
| *5-The research environment in which the process took place was described* | [21, 24–29, 31–37] | 14 (82.3) |
| *6-The political environment in which the process took place was described* | [29] | 1 (5.8) |
| *7-The economic/financial environment in which the process took place was described* | [32] | 1 (5.8) |
| **Use of a comprehensive approach** | | |
| *8-The process of priority setting was described in detail* | [22, 23, 29, 31–33] | 6 (35.2) |
| **Inclusiveness** | | |
| *9-The participants involved in setting research priorities were described* | [21–23, 26, 28–37] | 14 (82.3) |
| *10-An appropriate representation of expertise was included* | [22, 23, 28, 30–33, 36] | 8 (47) |
| *11-An appropriate representation of sex was included* | [22, 23, 33, 34, 36] | 5 (29.4) |
| *12-An appropriate representation of regional participation was included* | [21–23, 28–31, 33, 34, 36, 37] | 11 (64.7) |
| *13-Relevant health sectors and other constituencies were included* | [22, 23, 26, 28–30, 32, 36, 37] | 9 (52.9) |
| **Information gathering** | | |
| *14-The information and sources used to inform the priority setting exercise were referenced* | [21–33, 37] | 14 (82.3) |
| **Planning for implementation** | | |
| *15-Plans for translation of research priorities were discussed* | [21, 31, 37] | 3 (17.6) |
| *16-Who will implement the research priorities and how* | [21, 31, 37] | 3 (17.6) |
| **Criteria** | | |
| *17-Relevant criteria to focus discussion around setting priorities were stated* | [22, 23, 26, 32, 33, 37] | 6 (35.2) |
| **Methods for deciding on priorities** | | |
| *18-Approach for deciding on priorities was described (e.g., consensus or metrics based* | [21–27, 29–33, 37] | 13 (76.4) |
| **Evaluation** | | |
| *19-When and how evaluation of the established priorities and the priority setting process will take place were defined (e.g., multiple sessions)* | 0 | 0 |
| **Transparency** | | |
| *20-Clarity about the approach used was stated, i.e., who set the priorities how priorities were set* | [21–23, 25–32, 36, 37] | 13 (76.4) |

they used [22, 23, 31]. These included a detailed discussion around an interactive device used for the process [22, 23] and the materials used for each rounds of the process (e.g., flip chart paper, markers; audio recorders). The health environment was described by nearly all studies aside from [22, 23, 34] and the research environment was described by all studies excluding [22, 23, 30] yet only one study described the political environment in which the prioritisation exercise occurred [29]. Similarly, only one study described the economic environment context [32].

**Theme 2: Use of a comprehensive approach.** Six studies identified using comprehensive frameworks to conduct their research priority setting exercises [22, 23, 29, 31–33]. Four of

them modified existing frameworks. One from the Child Health and Nutrition Research Initiative [32]; one from the James Lind Alliance [33]; two adapted an existing deliberation exercise: CHAT framework [22, 23]. Two studies developed comprehensive protocols and used them to guide their research priority setting exercises [29, 31]. The remaining eleven studies did not report using an established framework to guide the process.

**Theme 3: Inclusiveness.**  Overall, most studies described the participants involved in the process. However, some studies provided more demographic information than others, including age, ethnicity, sex, and occupation of participants. Two studies did not provide any demographic details [24, 27]. The inclusion of a diverse range of key technical experts such as policy makers, service providers, academics, researchers and health care practitioners was described in most cases and these groups formed a core working group/council in the first stage of the prioritisation exercise in some studies [22, 26, 29, 30, 32]. These experts either had experience of working with the groups they were setting priorities for, or extensive knowledge of the groups (according to the authors). There was PPI in the research team albeit in fewer instances [29, 30].

Underpinning most of the prioritisation exercises were strong notions of equality, fairness and justice for people underrepresented in research, with a focus on community research ownership, community action, collaboration and partnership. An appropriate level of BAME PPI involvement was included in most studies [22, 23, 28–31, 33, 34, 36, 37]. Six studies did not report any BAME PPI [21, 24, 25, 27, 32, 35] and one study used a very small sample of BAME PPI [26].

Of the ten studies that mentioned the sex of participants, half contained a disproportionately larger number of female participants. However, it was noted that females are underrepresented in research more generally, so over-representation was not deemed an issue under this circumstance. The majority of studies reported an appropriate representation of regional participation and had included relevant sectors in their priority setting process.

**Theme 4: Information gathering.**  The majority of studies used a working group or planning committee consisting of experts in the field, in order to define domains and categories in which to direct the research priorities from the beginning of the exercise. Most studies reported the range of technical data required to inform the discussion on research priorities ranging from presentations, literature reviews, informal and formal discussions, conferences, health assessments and surveys. Initial documentation containing stakeholder priorities and research priorities set by external bodies in the field were used in some instances to compare previous research priorities and those identified in the exercise. Consultation with BAME community stakeholders prior to the studies was reported in a minority of studies [25, 31, 33].

**Theme 5: Planning for implementation.**  A limited number of studies reported plans to convert research priorities into projects. One included the prioritised list in a request for proposals to address issues around pre-term birth [31]. Another disclosed that workshops and targeted seminars were developed, along with a grant application submission, and sponsored projects to reduce childhood obesity [21] and several projects in the community were established to aimed at obesity reduction in Latino children as a result of their research priority setting study [37]. Plans for pilot studies were also established from research agendas [21, 31, 37].

**Theme 6: Criteria.**  Six studies stated that criteria had been chosen to focus the discussions [22, 23, 26, 32, 33, 37] yet criteria was only made explicit by [33] who cited; burden of illness, inequalities, cost to the NHS, and impact on family and child as criteria for topic submission, and [32] who listed; answerability, significance, and practical application, as criteria to set priorities.

**Theme 7: Methods for deciding on priorities.**  Studies adopted either a consensus-based approach, [24, 27, 29, 33] a metrics-based approach [21, 26, 28, 30, 37] or a combination of

both [22, 23, 25, 31, 32]. A small minority of studies used thematic analysis to identify themes borne from focus group discussions and described these as the research priorities [34–36]. Ranking and/or consensus was not used to determine these priorities. Likert scales were the most popular ranking method used for prioritization. Dot voting was used to rank in one study [31]. Surveys, interviews (phone and face to face), focus groups, the Delphi method (adapted and original), the nominal group technique (adapted and original), were identified as common methods for deciding on priorities. One study did not disclose its methods for deciding on priorities [24].

**Theme 8: Evaluation.**   No studies reported plans to update research priorities. Despite this, in a small number of studies, researchers did go on to conduct further research priority exercises for the same ethnic population they studied in an earlier research priority study [22, 25, 35], albeit with the aim of more diverse stakeholder inclusion [23, 34] and to establish a research agenda in a different topic area [26]. The most recent study included in this review expressed a desire to conduct a similar study with other ethnic groups [30].

**Theme 9: Transparency.**   Most studies were explicit about who set the priorities by providing information on participant characteristics. Studies were also transparent in describing the stages of the process that involved different stakeholders. For instance, initial topic areas may have been identified by a core group; community/patients/service users, or by professional stakeholders, to be ranked in the final stage of the process by another group of stakeholders. Most studies detailed how they set priorities; however, the extent to which an explanation on how priorities were set varied across studies.

Some studies made transcripts available [31], as well as reports [25] along with an interim report which included the research plan, research tools and a list of key informants [29] and finally, a consensus statement [24]. Few studies provided an evaluation of their priority setting process [22, 25, 26, 30, 31, 33]. Examples included: an acknowledgment of little to low involvement from BAME communities in setting priorities either at any given stage of the exercise [25, 26, 30]; evaluation of the information gathering stage as well as the plausibility of incentivizing responses to increase response rates by expert stakeholders [33]; discussion around data collection about ethnicity encountered technical difficulties which resulted in the loss of data for 25% of the participants, which could have markedly influenced the study findings [22].

## Discussion

This review provides an overview of research priority setting within studies for BAME health. Publications in a range of different topic areas were identified such as knee surgery, mental health, preterm birth, and child obesity. Identified priorities include the design and delivery and location of mental health services for refugees; research into the effectiveness of alternative medicines for South Asian children; cancer prevention and early detection in indigenous Australians, and integration of behavioural and cultural components into research to prevent and treat paediatric obesity in predominantly Black and Hispanic communities. By applying a checklist of good practice in research priority setting by Viergever et al. [17] to the prioritisation exercises, a number of strengths and weaknesses were identified which influenced the quality of the research prioritisation exercises within the studies. None of the studies fulfilled all the good practice criteria. This suggests there remains significant work to do to achieve effective research priority setting in BAME health.

Our findings suggest that the greatest failure of studies when assessed against the good practice checklist, concerned the criteria *evaluation*. The majority of studies presented procedures and outcomes which were said to inform and assist funders and policy makers on making better decisions and assist researchers in doing more work in the area, especially in involving

BAME populations in setting research priorities. Yet very few exercises made explicit any plans to translate the priorities into projects. It should be noted that this issue is not limited to research prioritisation for BAME health only. Prioritisation literature has demonstrated that lack of evaluation of outcomes is a common problem across research prioritisation studies more widely [14]. However, this has been further identified as a barrier to BAME participation in research. It is perceived that there would be no personal benefit of participating [38] and it is a common complaint from BAME communities that they are not informed of the outcomes of a study after taking part [39]. Thus, it is very important to disseminate results back to the community and participants involved so they can be assured that their views were in fact incorporated and their involvement was not tokenistic. Strategies to promote outcome evaluations in BAME population research could include reporting the acceptability of the exercise to those involved in the process [14] and performing an impact analysis, for example, as a review of research performed can be valuable in that it can enforce discussion on issues surrounding implementation [17].

None of the studies used the original versions of established research priority setting approaches, recognizing that they were unsuitable for involving BAME participants. For instance, some studies advised against using well known comprehensive frameworks where BAME patient and public stakeholders co-produced research priorities alongside professional stakeholders, noting that it may lead to the possibility of further muting seldom heard voices, instead, they either adapted comprehensive approaches where opinions could be expressed freely, or developed their own, deeming them more inclusive [29, 31, 33].

It is widely recognised that community engagement in priority setting is a key means for setting research questions and topics that are relevant and beneficial to them. Yet, without addressing power dynamics, their engagement can be tokenistic [40]. The studies in this review that developed their own framework, focussed on strong notions of patient/community-oriented research, co-production, community engagement, and BAME research ownership, and cited the importance of participatory approaches to research. They were especially mindful of involving BAME participants in setting priorities and described a more thorough recruitment strategy to fulfil this objective. Examples included examining potential barriers to recruitment and steps taken to overcome them such as arranging transportation and accommodation of issues around language and low literacy.

One particular area where some studies were lacking was the appropriate involvement of BAME communities in establishing research priorities. Sample sizes of BAME groups were either too small or there was no involvement of these groups at all, in any stage of the process. Discourse has established that marginalized communities are often excluded from priority setting exercises due to issues surrounding language barriers and difficulties accessing communities [41]. Given that research findings show that BAME groups react favourably and show a willingness to be involved in research, perceiving research participation as an opportunity or even a right [42], this is a missed opportunity. As well as this, ignoring BAME concerns has generated scepticism and community anger towards health research, especially among BAME groups [43].

Interestingly, many of the studies that scored highly in quality, had involved a high number of BAME research participants in various stages of their prioritisation exercise [22, 29, 31, 33, 37]. Some of them had developed novel frameworks to conduct research priority setting specifically for BAME groups and recommended that their framework be used by other researchers aiming to set research priority agendas in BAME health. Involving BAME communities in each stage of the process was deemed fundamental in their studies. For instance, they ran pilot studies prior to the prioritisation process or enlisted members of the community to provide feedback on the topics guide used to generate discussion around priorities. They made efforts

to build trust and facilitate smooth participation in their exercises. This was done in various ways such as enlisting community facilitators to guide discussions, utilising bilingual interpreters, recruiting community leaders in a position of trust in their communities, organizing the exercises in locations familiar to participants, holding focus groups specifically for different cultural groups, and running women only groups. In the literature, these have all been identified as effective ways of facilitating BAME community participation in research [38, 44]. This is in line with the UK's National institute for health research INVOLVE guidance [45] on co-producing a research project that emphasizes embracing diversity and the development of structures and practices to allow for the involvement of all stakeholders required for a project. There is also an argument that experiential knowledge from those directly affected by the issue being researched improved the quality and relevance of the research such as identifying appropriate research questions and improving the clarity of communications [46].

With regards to applying *criteria* to generate research priorities, only two studies explicitly stated the criteria to guide the process [32, 33] it was evident that an ethical framework supported the process of priority setting in most of the studies as justice and fairness were explicitly mentioned by researchers when describing the context of the proposed research. However, this may not be sufficient given that explicitly defined criteria is particularly important in that it could provide justification to satisfy funders and policy makers so they may fund, support and utilise the priorities [47, 48].

## Strengths and limitations

To our knowledge, this is the first systematic review to characterise and evaluate published research priority setting studies relevant to the health of BAME populations. The strengths of this review include the description of reported health research priorities in BAME populations, and the application of a comprehensive methodological framework to evaluate processes. However, the review is not without its limitations. For example, the search was limited to include studies in the English language only, which could have excluded research priority studies in other languages.

## Implications and recommendations for future research

Since the publication of the narrative review from Bryant et al. (2014) and according to the results from this review, barely any improvements have been made in terms of evaluation from prioritization processes in exercises. Quality research that adheres to good practice guidelines are required to make a difference to BAME communities and improve outcomes, thus helping close the health inequality gap amongst BAME communities. Identifying potential barriers to recruiting BAME communities in research priority setting and putting measures in place to overcome these, could be very useful in increasing their involvement. This ensures relevance of the research to meet their needs and address health inequalities.

## Supporting information

**S1 File. Study PRISMA checklist.**
(DOC)

## Author Contributions

**Conceptualization:** Halima Iqbal, Jane West, Melanie Haith-Cooper, Rosemary R. C. McEachan.

**Data curation:** Halima Iqbal.

**Formal analysis:** Halima Iqbal, Melanie Haith-Cooper.

**Investigation:** Halima Iqbal.

**Methodology:** Halima Iqbal.

**Project administration:** Halima Iqbal.

**Resources:** Halima Iqbal.

**Software:** Halima Iqbal.

**Supervision:** Halima Iqbal.

**Validation:** Halima Iqbal.

**Visualization:** Halima Iqbal.

**Writing – original draft:** Halima Iqbal.

**Writing – review & editing:** Halima Iqbal, Jane West, Melanie Haith-Cooper, Rosemary R. C. McEachan.

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
