## [Decision Letter · Decision Letter 0]

12 Mar 2021

PONE-D-20-29158

Research priority setting in Black, Asian and minority ethnic health: A systematic review

PLOS ONE

Dear Dr. Iqbal,

Thank you for submitting your manuscript to PLOS ONE. After careful consideration, we feel that it has merit but does not fully meet PLOS ONE’s publication criteria as it currently stands. Therefore, we invite you to submit a revised version of the manuscript that addresses the points raised during the review process.

We look forward to receiving your revised manuscript.

Kind regards,

Baltica Cabieses, PhD

Academic Editor

PLOS ONE

Journal Requirements:

2. Please specify the time frame for your search - i.e., did you search for articles from the inception of the databases? In addition, please ensure you include all relevant articles published to date.

3. Please state how many authors participated in the article search.

4. Please ensure that you include a title page within your main document. You should list all authors and all affiliations as per our author instructions and clearly indicate the corresponding author.

5. We note you have included a table to which you do not refer in the text of your manuscript. Please ensure that you refer to Table 1 and 2 in your text; if accepted, production will need this reference to link the reader to the Table.

7. We note that Figure 1 in S5 includes an image of a participant in the study. 

Reviewers' comments:

Reviewer's Responses to Questions

**Comments to the Author**

1. Is the manuscript technically sound, and do the data support the conclusions?

Reviewer #1: Yes

Reviewer #2: Yes

2. Has the statistical analysis been performed appropriately and rigorously? 

Reviewer #1: Yes

Reviewer #2: N/A

3. Have the authors made all data underlying the findings in their manuscript fully available?

Reviewer #1: Yes

Reviewer #2: Yes

4. Is the manuscript presented in an intelligible fashion and written in standard English?

Reviewer #1: Yes

Reviewer #2: Yes

5. Review Comments to the Author

Reviewer #1: The systematic review “Research priority setting in Black, Asian and minority ethnic health” well executed and scientifically founded study. Research question, hypothesis, methods and results are clearly described.

Reviewer #2: The manuscript gives an interesting critical view of the evidence on research priority setting in BAME population. The strengths of this study include the application of a comprehensive methodological framework to evaluate processes. As well as reporting the non-compliance to good practice guidelines. However, there are missing details in the methodology to ensure the standards of the PRISMA statement. Besides description of the characteristics and main results of included studies, since the reader might be interested in having more information of priority research topics in BAME populations, beyond the quality of these studies. Furthermore, there are punctuation errors that must be revised throughout the document.

Title

It might be necessary to specify the scope of the review, considering that the authors proposed: “to determine the extent to which the evidence followed good practice principles”.

Abstract

Page 1. I suggest including a brief description of search terms and inclusion criteria.

Page 2. The results section does not report the priority research topics (mental health, preterm birth, child obesity etc.) found in this study. Moreover, I suggest highlighting the concern related to evaluation criteria, as stated in the main text.

Introduction

Page 4. Although the authors have mentioned evidence on priority setting methods and the increase of priority setting exercise, there is no cited literature related to the specific context of BAME population research. Therefore, I recommend to clearly state the knowledge gap in BAME population and provide reference for “given the stark increase in research priority setting in the past decade, along with an increase of discourse around evaluation of research priority setting initiatives”.

Methods

Page 6. Regarding the study selection, I suggest specifying characteristics of the admitted population, study types and a well stablished main outcome. The latter is necessary to understand the applied criteria by the authors and to justify their selections. Therefore, it will support the statement “This included studies where an ethnically diverse population was described in order to determine how priorities may differ between ethnic groups”. Table 1. I recommend revising the description of inclusion and exclusion criteria in order to make a clear parallel between them.

Page 7. Although the quality appraisal tool is presented in detail, there is some missing information related to the quality score interpretation (table 3 only shows percentages based on meet criteria).

Page 9. There is a description of selection process that should be included before the quality appraisal. Furthermore, there is no information of data collection and synthesis (data extraction procedures and qualitative analysis). I suggest revising and follow a consistent structure with the PRISMA statement.

Results

Page 11. The first paragraph of the result section does not include a brief description of the characteristics (participants when possible, study type etc.) and main outcomes of included studies. This section must inform the readers of what is known about the research priority topics to highlight the current needs in BAME population. The results are focused on the quality criteria, instead of providing a comprehensive analysis of all the data that has been collected.

Table 3. I suggest including columns for participant’s description and main outcomes related to the topic and scope of each study. Furthermore, the quality score column might include a brief explanation, so the reader will identify the flaws of each study. Please Define E-health at the table legend.

Discussion

Page 19. The first paragraph of the discussion must be accompanied with a brief critical synthesis of priority topics, beyond just mentioning them.

Page 20. I suggest citing o propose strategies to promote evaluation of outcomes in BAME population research.

Page 21. The authors state “sample sizes of BAME groups were either too small or there was no involvement of these groups at all, in any stage of the process”, which raise questions about the selection criteria; since the reader might be expecting studies that include BAME population through the process.

Page 23. The Last paragraph of the discussion does not highlight the concern related to evaluation of the process. Please report a summary of study strengths and limitations

6. PLOS authors have the option to publish the peer review history of their article (what does this mean?). If published, this will include your full peer review and any attached files.

Reviewer #1: No

Reviewer #2: No

---

## [Author Response · Author response to Decision Letter 0]

15 Apr 2021

Academic editor’s comments 

1. Please ensure that your manuscript meets PLOS ONE's style requirements, including those for file naming 

Author - The style changes have been made throughout the manuscript, including naming

2. Please specify the time frame for your search - i.e., did you search for articles from the inception of the databases? In addition, please ensure you include all relevant articles published to date

Author - The following sentence has been added “we searched databases from their inception to July 2020” (p.7)

3. Please state how many authors participated in the article search

Author - The following sentence has been added “the principal researcher (HI) independently conducted the article search” (p.6)

4. Please ensure that you include a title page within your main document. You should list all authors and all affiliations as per our author instructions and clearly indicate the corresponding author 

Author - A title page has been added and has adhered to PLOS1 author guidelines

5. We note you have included a table to which you do not refer in the text of your manuscript. Please ensure that you refer to Table 1 and 2 in your text; if accepted, production will need this reference to link the reader to the Table 

Author - Table 1 is now referred to in the text on p.8. Table 2 is referred to on p.10

6. Please include captions for your Supporting Information files at the end of your manuscript, and update any in-text citations to match accordingly. Please see our Supporting Information guidelines

Author - The supporting Information file now includes captions which can be found at the end of the manuscript. In text-citations have been updated.

7. We note that Figure 1 in S5 includes an image of a participant in the study. 

If you are unable to obtain consent from the subject of the photograph, you will need to remove the figure and any other textual identifying information or case descriptions for this individual 

Author - Studies included in the review have been removed from the supporting information files. Included studies can be located in the reference list. Therefore, S5 no longer exists with the image requested to be removed 

Reviewer 1 comments 

Grammar errors on page 6, 15, 16, 20, 21, 22, 23 

Author - The errors have been corrected

Since in the introduction the example of health conditions from UK Black African women was mentioned, the question rises why the search term “Africans” was not included in the search, to increase the reach of articles including this specific sub-group - Page 4. The example given of health conditions among UK Black African women has been removed 

Introduction: “This systematic review is interested to see if there has been any progress in research priority process evaluation since then, with a specific focus on BAME health, given the stark increase in research priority setting in the past decade, along with an increase of discourse around evaluation of research priority setting initiatives” � Recommendation to include the conclusion that since the publication of the narrative review from Bryant et al. (2014) and according to the results from this review (none of the studies fulfilled all the good practice criteria) barely any improvements have been made in terms of evaluation from prioritizations processes in exercises 

Author - Page 42. The recommendation has been added to the section Implications and recommendations for future research 

Reviewer 2 comments 

Title

It might be necessary to specify the scope of the review, considering that the authors proposed: “to determine the extent to which the evidence followed good practice principles”. Author - Title has been changed to “A systematic review to identify research priority setting in Black and minority ethnic health and evaluate their processes” 

Abstract

Page 1. I suggest including a brief description of search terms and inclusion criteria - 

Author - page 2. A description of search terms and inclusion criteria have been added to the abstract 

Page 2. The results section does not report the priority research topics (mental health, preterm birth, child obesity etc.) found in this study. Moreover, I suggest highlighting the concern related to evaluation criteria, as stated in the main text 

Author - Page 2. The results section has been amended to include the different topic areas found in the study. I have added in a sentence highlighting the concern related to the evaluation, which now states “None of the exercises followed all the good practice principles as outlined in the checklist"

Introduction

Page 4. Although the authors have mentioned evidence on priority setting methods and the increase of priority setting exercise, there is no cited literature related to the specific context of BAME population research. Therefore, I recommend to clearly state the knowledge gap in BAME population and provide reference for “given the stark increase in research priority setting in the past decade, along with an increase of discourse around evaluation of research priority setting initiatives” 

Author - Page 5. The following sentence has been added to address the reviewer’s concern: “It is also unclear whether research prioritisation exercises have been undertaken for BAME health” Page 5-6. References have been provided for the statement quoted by the reviewer 

Methods

Page 6. Regarding the study selection, I suggest specifying characteristics of the admitted population, study types and a well stablished main outcome. The latter is necessary to understand the applied criteria by the authors and to justify their selections. Therefore, it will support the statement “This included studies where an ethnically diverse population was described in order to determine how priorities may differ between ethnic groups”. Table 1. I recommend revising the description of inclusion and exclusion criteria in order to make a clear parallel between them 

Author - Table 1 has been amended to make this distinction clear

Page 7. Although the quality appraisal tool is presented in detail, there is some missing information related to the quality score interpretation (table 3 only shows percentages based on meet criteria) 

Author - page 12. the information has been added under the section data collection

Page 9. There is a description of selection process that should be included before the quality appraisal. Furthermore, there is no information of data collection and synthesis (data extraction procedures and qualitative analysis). I suggest revising and follow a consistent structure with the PRISMA statement 

Author - Page 7 -8. The section has been amended to include this information 

Page 12. Information on data collection and synthesis has been added 

Results

Page 11. The first paragraph of the result section does not include a brief description of the characteristics (participants when possible, study type etc.) and main outcomes of included studies. This section must inform the readers of what is known about the research priority topics to highlight the current needs in BAME population. The results are focused on the quality criteria, instead of providing a comprehensive analysis of all the data that has been collected.

Author - Page 14. The first paragraph of the results section now includes a brief description of the participants in each study and the methods used in studies. Although Table 3 now includes a column displaying the main outcomes (research priorities) for each study, as suggested by the reviewer, multiple research priorities were identified throughout the studies, in different topic areas, as is typical in priority setting exercises as they encompass a wide spectrum of views. As such, we are unable to provide a comprehensive analysis of the priorities themselves in the first paragraph of the results section. Instead, we listed the top identified priority in each study. If the reader would like to view the full list of priorities for each study, the studies can be found in the reference list.

Table 3. I suggest including columns for participant’s description and main outcomes related to the topic and scope of each study. Furthermore, the quality score column might include a brief explanation, so the reader will identify the flaws of each study 

Author - Columns have been added displaying the population and the main outcome of each study. We have added information in the quality score column, in the first box and signposted to Table 2 which provides detail of the quality criteria

Please Define E-health at the table legend

Author - Page 27. health has been defined at the table legend

Discussion

Page 19. The first paragraph of the discussion must be accompanied with a brief critical synthesis of priority topics, beyond just mentioning them.

Author - Page 37. Examples of priority topics have now been given 

Page 20. I suggest citing o propose strategies to promote evaluation of outcomes in BAME population research

Author - Page 38-39. Strategies have been proposed 

Page 21. The authors state “sample sizes of BAME groups were either too small or there was no involvement of these groups at all, in any stage of the process”, which raise questions about the selection criteria; since the reader might be expecting studies that include BAME population through the process

Author - The selection criteria have been amended and it is now clear that BAME participation in studies was not a pre-requisite for study inclusion

Page 23. The Last paragraph of the discussion does not highlight the concern related to evaluation of the process. Please report a summary of study strengths and limitations

Author - Page 42. The concern regarding evaluation of the process is now highlighted

Page 42. Strengths and limitations of the study have been provided

---

## [Editor Report · Decision Letter 1]

3 May 2021

A systematic review to identify research priority setting in Black and minority ethnic health and evaluate their processes

PONE-D-20-29158R1

Dear Dr. Iqbal,

We’re pleased to inform you that your manuscript has been judged scientifically suitable for publication and will be formally accepted for publication once it meets all outstanding technical requirements.

Kind regards,

Baltica Cabieses, PhD

Academic Editor

PLOS ONE

---

## [Editor Report · Acceptance letter]

19 May 2021

PONE-D-20-29158R1 

A systematic review to identify research priority setting in Black and minority ethnic health and evaluate their processes 

Dear Dr. Iqbal:

I'm pleased to inform you that your manuscript has been deemed suitable for publication in PLOS ONE. Congratulations! Your manuscript is now with our production department. 

Kind regards, 

on behalf of

Dr. Baltica Cabieses 

Academic Editor

PLOS ONE